# Treatment of Antihypertensive and Cardiovascular Drugs in Supercritical Water: An Experimental and Modeled Approach

Isabela M. Dias [1], Lucas C. Mourão [1], Guilherme B. M. De Souza [1], Jose M. Abelleira-Pereira [2], Julles M. Dos Santos-Junior [3], Antônio C. D. De Freitas [4], Lucio Cardozo-Filho [5], Christian G. Alonso [1] and Reginaldo Guirardello [3],*

1   Instituto de Química, Universidade Federal de Goiás (UFG), Av. Esperança s/n, Campus Samambaia, Goiânia 74690-900, GO, Brazil; isabelamilhomem@ufg.br (I.M.D.); lucasclementinom@gmail.com (L.C.M.); guilherme_botelho@ufg.br (G.B.M.D.S.); christian@ufg.br (C.G.A.)
2   Department of Chemical Engineering and Food Technology, Faculty of Sciences, University of Cádiz, International Excellence Agrifood Campus (CeiA3), Puerto Real, 11510 Cádiz, Spain; jose.abelleira@uca.es
3   School of Chemical Engineering, University of Campinas (UNICAMP), Av. Albert Einstein 500, Campinas 13083-852, SP, Brazil; jullesmitoura7@gmail.com
4   Engineering Department, Exact Sciences and Technology Center, Federal University of Maranhão (UFMA), Av. Dos Portugueses, 1966, Bacanga, São Luís 65080-805, MA, Brazil; tonyfrt12@gmail.com
5   Programa de Pós-Graduação em Engenharia Química, Universidade Estadual de Maringá (UEM), Avenida Colombo, 5790–Zona 7, Maringá 87020-900, PR, Brazil; lcfilho@uem.br
*   Correspondence: guira@unicamp.br

**Abstract:** Pharmaceutical pollutants are considered emerging contaminants, representing a significant concern to the ecosystem. Thus, this study reports on the degradation of antihypertensive and cardiovascular drugs (atenolol, captopril, propranolol hydrochloride, diosmin, hesperidin, losartan potassium, hydrochlorothiazide, and trimetazidine) present in simulated wastewater through applying the technology of oxidation using supercritical water (SCW). The operational parameters of the treatment process, particularly the feed flow rate, temperature, and concentration of $H_2O_2$, were assessed. A central composite design of experiments associated with differential evolution was employed in the optimization. Both liquid and gaseous phase products were submitted to physical–chemical characterization. As a result, the optimized conditions for the treatment were discovered to be a feed flow rate of 13.3 mL/min, a temperature of 600 °C, and a $H_2O_2$ oxidation coefficient of 0.65, corresponding to the oxygen stoichiometric coefficient in the carbon oxidation chemical reaction. Under optimal conditions, the total organic carbon (TOC) decreased from 332 to 25 mg/L (92.1%), and the pharmaceutical molecules underwent near-complete degradation. The physical–chemical parameters also met with the main environmental regulations for wastewater disposal. The compounds determined in the gaseous phase were $CO_2$ (97.9%), $H_2$ (1.3%), $CH_4$ (0.3%), and CO (0.5%.). Additionally, a modeling thermodynamic equilibrium of the system was performed, based on the experimental data. The results revealed that SCW technology has a great potential to oxidize/degrade organic matter and can be applied to treat pharmaceutical pollutants.

**Keywords:** supercritical water treatment; emergent pollutants; pharmaceutical pollutants; design of experiments; thermodynamic equilibrium simulation

## 1. Introduction

Several classes of pharmaceutical molecules have been identified in the environment. Consequently, these substances are now classified as emerging contaminants, capable of persisting in various environmental matrices [1,2]. As a result, wastewater treatments are essential to remove such potentially toxic compounds. However, traditional methods of treatment are not efficient since wastewater treatment plants were not initially designed to eliminate emerging pollutants [3]. Cardiovascular diseases and hypertension represent

significant global health issues, necessitating the use of antihypertensive medications. These medications play a crucial role in reducing cardiovascular morbidity, and, consequently, decreasing mortality [4]. However, the presence of such medications in water bodies has gained attention, due to their potentially negative effects on aquatic organisms and the disruption that they can cause to the equilibrium of the ecosystem [5].

Antihypertensive and cardiovascular medications act as the primary methods of therapeutic interventions to stabilize irregular cardiac rhythms and regulate blood pressure [6]. In 2004, hypertension was directly responsible for 12.8% of global deaths, and it is still a major cardiovascular risk factor with a significant impact on mortality [7,8]. Furthermore, according to the World Health Organization, approximately 1.5 billion people worldwide have hypertension, a number expected to rise, which has led to a continued increase in the prescription and consumption of antihypertensive drugs [9]. Given the chronic nature of these conditions, long-term medical treatment becomes a necessity. In the human body, most of these drugs undergo hepatic metabolism and renal excretion, resulting in the release of a considerable quantity of unchanged drugs or by-products into the environment [10]. The presence of antihypertensive drug molecules in municipal wastewater has been observed in several countries. Subedi and Kannan (2015) monitored two centralized wastewater treatment plants in the Albany area (New York, NY, USA) and determined that the daily dose per thousand inhabitants for atenolol, propranolol, diltiazem, and verapamil was 316, 50.7, 45.8, and 30 mg, respectively [11]. In the Tagus Estuary (Portugal), the occurrence of antihypertensive drugs, including indapamide (1.09–4.67 ng/L), irbesartan (7.57–161.9 ng/L), and losartan (1.52–64.7 ng/L), as well as β-blockers such as atenolol (0.49–0.49 ng/L), bisoprolol (0.02–4.66 ng/L), carvedilol (0.53–1.01 ng/L), and propranolol (0.02–1.89 ng/L), was also observed [12].

The efficacy and the efficiency of various treatment methods, such as adsorption, ion exchange, ozonation, membrane separation, electro-oxidative processes, and advanced oxidative processes, have been evaluated for the degradation or removal of contaminants. However, it is important to note that these methods, when employed individually, may not be sufficient to adequately degrade or remove these contaminants [5,13–15]. Considering the need to treat recalcitrant contaminants, it becomes necessary to enhance the treatment processes to achieve effective results. In this regard, the application of supercritical water (SCW) technology can be a promising alternative for treating wastewater containing cardiovascular and antihypertensive drug molecules. Unlike most of the conventional treatments that are not efficient enough or may show limitations for large-scale application, SCW treatment shows the potential to be highly oxidative, which allows the fast and efficient gasification of organic molecules.

The SCW oxidation process consists of the use of water above its critical point (with a pressure and temperature above 22.12 MPa and 374.15 °C, respectively) as an oxidative medium for the treatment of organic matter. This phenomenon is possible due to the distinctive characteristics that are exhibited by water under supercritical conditions. Under these conditions, water undergoes significant reductions in viscosity, the dielectric constant, density, and the ionic product, resulting in a reduced polarity and heightened solubility for the organic molecules. Thus, the breakdown of the molecules into smaller ones occurs through multiple pathways and fast reactions [16,17]. This particular technology has had successful applications in various fields, including the treatment of wastewater [18], industrial effluents [19], and hospital effluents [20], among others. Nonetheless, the use of SCW processes for the degradation of antihypertensive and cardiovascular molecules in water has not been investigated in depth as far as the authors of this work are aware. Hence, certain aspects can be enhanced and thoroughly assessed, including drug degradation rates, the degradability of total organic carbon and chemical oxygen demand, the reaction time, and the characterization of the gas phase generated during the treatment process.

Herein, given these gaps, the continuous flow supercritical water process was evaluated and applied to treat simulated wastewater containing a wide range of medications commonly used worldwide for the treatment of cardiovascular diseases and hypertension. The simultaneous degradation of eight target molecules was investigated, namely atenolol, captopril, propranolol hydrochloride, diosmin, hesperidin, losartan potassium, hydrochlorothiazide, and trimetazidine. Furthermore, the effect of treatment process parameters, such as the temperature, $H_2O_2$ concentration, and feed flow rate, was assessed, as well as the behavior of the treatment process elucidation through the utilization of a simulation model of the thermodynamic equilibrium of the system.

## 2. Materials and Methods

### 2.1. Preparation of Simulated Wastewater

To prepare the simulated pharmaceutical wastewater, commercial pills of antihypertensive and cardiovascular medications were dissolved in 4 L of ultrapure water. Four pills with 50 mg of atenolol (EMS), four pills with 50 mg of captopril (GERMED), four pills with 40 mg of propranolol hydrochloride (NEOQUÍMICA), one pill with 35g of trimetazidine (SERVIER), four pills with 450 g of diosmin and 50 mg of hesperidin (NEOQUÍMICA), and four pills with 50 mg of losartan potassium and 12.5 mg of hydrochlorothiazide (LEGRAND) were dissolved in the water. The effect of the addition of $H_2O_2$ was also evaluated.

For this purpose, before the reaction test, $H_2O_2$ was added to the prepared solution in concentrations ranging from 0.07 to 2.93. This refers to the oxidation coefficient (OC), which is equivalent to the oxygen stoichiometric coefficient in the carbon oxidation chemical reaction. Thus, the *OC* was defined as shown in Equation (1):

$$OC = \frac{added\ oxygen\ amount}{theoretically\ required\ oxygen\ amount} \qquad (1)$$

The final solution had a total organic carbon (TOC) content of 332 mg/L. Furthermore, the feed solution was characterized as described in Section 2.4.

### 2.2. Supercritical Water (SCW) Reactor

The reaction tests were performed in a plug flow reactor fed with the simulated wastewater that had been prepared with the antihypertensive and cardiovascular medications. The experimental runs were performed in a reaction unit that comprised the equipment enumerated as follows: (1) a high-pressure pump; (2) a check valve; (3) a coiled pre-heater (heated by a vertical split tube furnace); (4) a tubular reactor (I.D.: 1.1 cm; L: 30 cm); (5) a vertical split tube furnace; (6) a jacketed coil condenser (coupled to a chiller); (7) a manometer; (8) a retention valve; (9) a safety relief; (10) a back pressure regulator (BPR); (11) a phase separator. FC and TC correspond to the flow controller and temperature controller, respectively. The liquid product flow rates were measured by the accumulation of fluid over time, which was then collected and stored, while the gas product flow rate was measured using a drum-type gas meter, as can be seen in Figure 1 that shows a representation of the reaction unit, as reported by Mourão et al. (2023) [21]. The process cold currents are represented by the color blue, while the warm currents are in red. The tubular reactor was made of Inconel (VRC 625), and the other components were made of stainless steel (SS 316L).

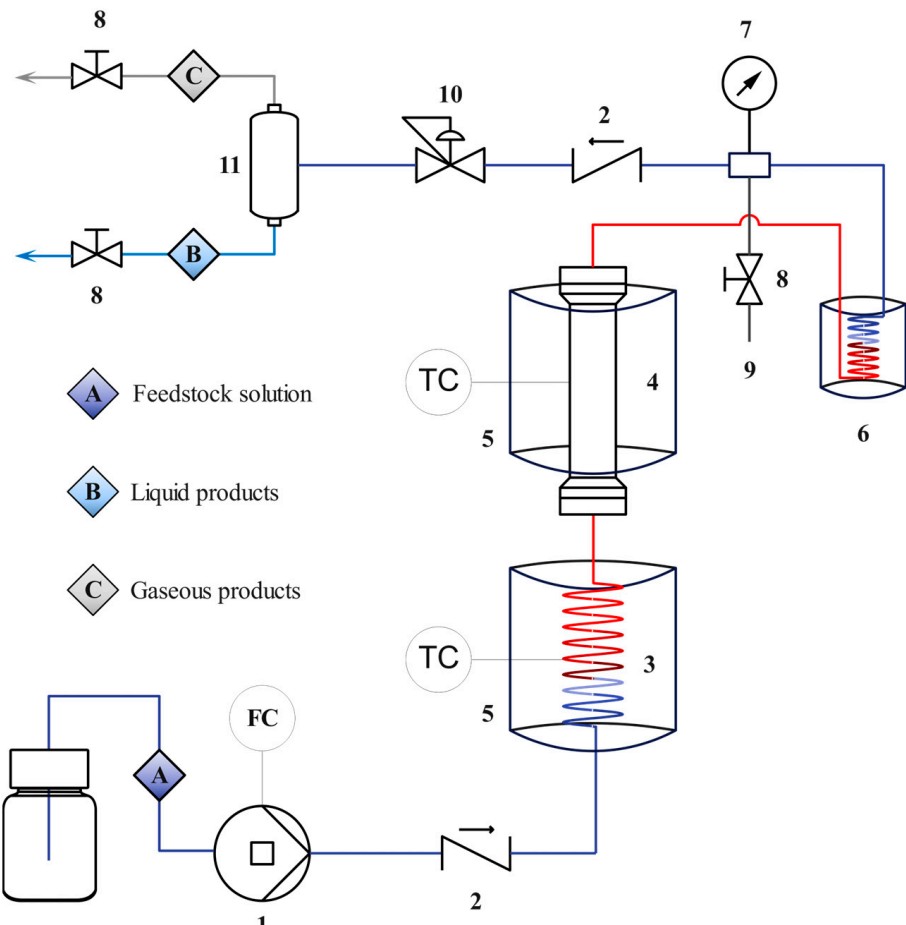

**Figure 1.** Schematic representation of the SCW reaction unit [21].

### 2.3. Reaction Tests' Conditions and Optimization

The reaction test parameters, including the temperature, the feed flow rate (that is, the reaction time), and the $H_2O_2$ concentration, were investigated. The pressure was kept constant (25 MPa) in all of the tests that were performed, since this parameter exhibits a negligible influence on the oxidation and/or partial oxidation of organic matter [21–23]. The removal of total organic carbon ($R_{TOC}$) was considered as the response parameter that was influenced by the change in the reaction conditions. The calculation of the $R_{TOC}$ was performed according to Equation (2). The TOC measurements obtained for the sample before and after processing in the plug flow reactor system are referred to as $TOC_{in}$ and $TOC_{out}$, respectively.

$$R_{TOC}(\%) = \left(1 - \frac{TOC_{out}}{TOC_{in}}\right) \times 100 \tag{2}$$

To determine the optimal experimental conditions, a central composite design methodology was used. This approach allowed the creation of a matrix where the experimental conditions were strategically placed at both the highest and lowest process limitation parameters. The limits of each parameter were defined on the basis of the operational capacity of the reaction system and previous studies [24]. The range of the parameters that were assessed ran from 407.2 to 692.8 °C for the temperature, from 6.6 to 23.4 mL/min for the feed flow rate, and from 0.07 to 2.93 for the *OC*, corresponding to the concentration of $H_2O_2$. Thereby, three factors (k = 3), six axial points (rotatability coefficient, $\alpha$ = 1.68), and four repetitions of the central point were considered. The experimental conditions are detailed in Table 1:

**Table 1.** Reaction conditions of experimental tests.

| Variables | Range | | | | |
|---|---|---|---|---|---|
| | −1.68 | −1 | 0 | 1 | 1.68 |
| Temperature of reactor (°C), factor T | 407 | 465 | 550 | 635 | 693 |
| Feed flow rate (mL/min), factor F | 6.6 | 10 | 15 | 20 | 23.4 |
| Concentration of $H_2O_2$, factor C | 0.07 | 0.65 | 1.5 | 2.35 | 2.93 |

| Test | Reactor temperature (°C) | Feed flow rate (mL/min) | [1] $H_2O_2$ (OC) | Reaction time (min) |
|---|---|---|---|---|
| 1 | 465 | 10 | 0.65 | 2.8 |
| 2 | 635 | 10 | 0.65 | 2.8 |
| 3 | 465 | 20 | 0.65 | 1.4 |
| 4 | 635 | 20 | 0.65 | 1.4 |
| 5 | 465 | 10 | 2.35 | 2.8 |
| 6 | 635 | 10 | 2.35 | 2.8 |
| 7 | 465 | 20 | 2.35 | 1.4 |
| 8 | 635 | 20 | 2.35 | 1.4 |
| 9 | 550 | 15 | 1.50 | 1.9 |
| 10 | 550 | 15 | 1.50 | 1.9 |
| 11 | 407 | 15 | 1.50 | 1.9 |
| 12 | 693 | 15 | 1.50 | 1.9 |
| 13 | 550 | 6.6 | 1.50 | 4.3 |
| 14 | 550 | 23.4 | 1.50 | 1.2 |
| 15 | 550 | 15 | 0.07 | 1.9 |
| 16 | 550 | 15 | 2.92 | 1.9 |
| 17 | 550 | 15 | 1.50 | 1.9 |
| 18 | 550 | 15 | 1.50 | 1.9 |

Note(s): [1] Stoichiometric-based oxidation coefficient calculated in reference to the initial TOC (332 mg C/L) concentration, which represents the amount required for the complete degradation of carbon.

The construction of the central composite design matrix and the analysis of variance (ANOVA) of the obtained data were performed using R software, version 4.3.0. The optimal condition for the degradation, taking into account the maximum organic matter partial gasification, was determined using a search algorithm based on the differential evolution method [25]. Using the Scilab 5.5.2 software, the classic method DE/rand/1/bin of the search algorithm—where DE = differential evolution, rand = random, 1 = number of vectors disturbed, and bin = binomial crossover—was applied, considering the following parameters: a population size (N) of 50 individuals, a disturbance rate (F) of 0.8, a crossing probability (Cr) of 0.8, and a stopping criterion based on reaching 250 generations.

### 2.4. Physical–Chemical Characterization

The samples were characterized both before and after processing under the SCW conditions. The analyzes were performed following the Standard Methods for the Examination of Water and Wastewater [26]. The following parameters were analyzed in the (i) liquid phase: the total organic carbon (TOC), a high-resolution mass spectrometry (HRMS), the biochemical oxygen demand (BOD), the chemical oxygen demand (COD), nitrite, nitrate, pH, and the metals determination; in the (ii) gaseous phase, gas chromatography was employed. The equipment and details of the methods are reported in the Supplementary Material.

### 2.5. Thermodynamic Evaluation

The equilibrium composition of a multi-component and multi-phase system was established through the minimization of Gibbs energy, at a constant pressure and constant temperature conditions. Equation (3), considering the mole quantities of each component within each phase, characterizes a system comprising gas, liquid, and solid phases [27]:

$$minG = \sum_{i=1}^{NC} n_i^g \mu_i^g + \sum_{i=1}^{NC} n_i^l \mu_i^l + \sum_{i=1}^{NC} n_i^s \mu_i^s \tag{3}$$

where $n_i$ and $\mu_i$ are the number of moles for each component and the chemical potential, respectively, while NC is the number of components. This equation is conditioned to two essential restrictions: (i) the non-negativity of the number of moles of each component in each phase (Equation (4)), and (ii) the balance of moles obtained by the atomic balance for reactive systems (Equation (5)):

$$n_i^g, n_i^l, n_i^s \geq 0 \tag{4}$$

$$\sum_{i=1}^{NC} a_{mi}(n_i^g + n_i^l + n_i^S) = \sum_{i=1}^{NC} a_{mi} n_i^0, m = 1, \ldots, NE \tag{5}$$

where g, l, and s represent the gas, liquid, and solid phases, $a_{mi}$ is the number of atoms of each element in a molecule, and NE is the number of types of atoms in the system, respectively. To construct the Gibbs energy minimization model in association with a SCW treatment system, the Gibbs energy has been minimized with the consideration that the components are only in the gas phase. Solid carbon ($C_{(s)}$) was considered as the unique compound in the solid phase. Equation (6) represents the Gibbs energy equation, with these considerations:

$$G = \sum_{i=1}^{NC} n_i^g \left( \mu_i^g + RT(\ln P + \ln y_i + \ln \phi_i) \right) + n_{C_{(s)}}^s \mu_{C_{(s)}}^0 \tag{6}$$

Non-ideality is represented by the fugacity coefficient, calculated by the virial equations of state truncated at the second virial coefficient. The second virial coefficient was calculated using the Pitzer correlation as modified by Tsonopoulos et al. (1979) [28,29]. The calculation of the fugacity coefficient is given in Equation (7):

$$\ln \hat{\phi}_i = \left[ 2\sum_{j}^{m} y_j B_{ij} - B \right] \frac{P}{RT} \tag{7}$$

In this equation, $\mu_i^g$ and $y_j$ are the chemical potential and the mole fraction of the component; R is the gas constant; T is the temperature of the system; P is the pressure; $\phi_i$ and ($\hat{\phi}_i$) are the fugacity coefficients of the pure component and the component in the mixture; m is the atom in a molecule; B is the second coefficient of the virial; and $B_{ij}$ is that cross second virial coefficient.

This combination of methods with a virial equation of state as the thermodynamic model was used in similar works reported in the literature and presented a good descriptive capacity. In study of Barros and co-workers, who evaluated the supercritical water gasification (SCWG) reaction of black liquor, and in Mitoura and co-workers who evaluated the thermal decomposition reaction of methane, we found examples of the use of this model with an excellent performance [30,31].

The presence of 23 chemical compounds in the outflow of the reaction system was considered, as can be seen in Table S1 of the Supplementary Material. These compounds were meticulously selected based on the experimental data acquired throughout the development of this paper. This list includes the primary gaseous compounds stemming from carbon, nitrogen, sulfur, oxygen, and hydrogen. Notably, certain compounds exist in ionic conjunction with sodium and potassium ions. However, these compounds were excluded from the scope of this thermodynamic model. This omission is due to the abundance of water employed within the system, rendering their inclusion impractical and unnecessary for the model's objectives.

The thermodynamic properties required to conduct a thermodynamic analysis of the reaction system, including the heat capacity, enthalpy, and Gibbs energy of formation, were obtained from the literature [32]. Table S2 shows the feed operating conditions (%wt of the reactants, the pressure, and the temperature range) that were required for the thermodynamic analysis of the SCW system. The thermodynamic analysis was performed using the Gibbs energy minimization methodology with the aid of the GAMS® 23.9.5

(General Algebraic Modeling System) software and the CONOPT 4 solver. The selected conditions aimed to represent the same experimental range studied in this work.

## 3. Results and Discussion

### 3.1. Reaction Tests' Assessment and Optimization of Operational Parameters

The application of the SCW technology for the degradation of the simulated wastewater containing antihypertensive and cardiovascular drugs was assessed. In the initial phase of this study, the influence of the reaction conditions, namely the temperature, the feed flow rate, and the concentration of $H_2O_2$, in the organic matter degradation was systematically investigated. The obtained experimental results of the $R_{TOC}$ for the tests described by the central composite design can be seen in Table 2.

**Table 2.** Results of the $R_{TOC}$ after treatment by the SCW process.

| Test | Reactor Temperature (°C) | Feed Flow Rate (mL/min) | [1] $H_2O_2$ (OC) | Reaction Time (min) | $R_{TOC}$ (%) |
|---|---|---|---|---|---|
| 1 | 465 | 10 | 0.65 | 2.8 | 22.86 |
| 2 | 635 | 10 | 0.65 | 2.8 | 49.47 |
| 3 | 465 | 20 | 0.65 | 1.4 | 27.37 |
| 4 | 635 | 20 | 0.65 | 1.4 | 50.14 |
| 5 | 465 | 10 | 2.35 | 2.8 | 84.42 |
| 6 | 635 | 10 | 2.35 | 2.8 | 98.49 |
| 7 | 465 | 20 | 2.35 | 1.4 | 71.33 |
| 8 | 635 | 20 | 2.35 | 1.4 | 88.78 |
| 9 | 550 | 15 | 1.50 | 1.9 | 96.32 |
| 10 | 550 | 15 | 1.50 | 1.9 | 97.35 |
| 11 | 407 | 15 | 1.50 | 1.9 | 59.75 |
| 12 | 693 | 15 | 1.50 | 1.9 | 95.17 |
| 13 | 550 | 6.6 | 1.50 | 4.3 | 97.94 |
| 14 | 550 | 23.4 | 1.50 | 1.2 | 79.77 |
| 15 | 550 | 15 | 0.07 | 1.9 | 9.15 |
| 16 | 550 | 15 | 2.92 | 1.9 | 94.13 |
| 17 | 550 | 15 | 1.50 | 1.9 | 95.96 |
| 18 | 550 | 15 | 1.50 | 1.9 | 93.51 |

Note(s): [1] Stoichiometric-based oxidation coefficient calculated in reference to the initial TOC (332 mg C/L) concentration, which represents the amount required for the complete degradation of carbon.

Moreover, the effects of the parameters can be easily comprehended through visualization using a response surface graph methodology. To that end, the independent variables were evaluated in relation to the total organic carbon removal function, which resulted in three graphical plots (see Figure 2).

Figure 2a shows that the temperature exhibited a more pronounced impact on the $R_{TOC}$ compared to the feed flow rate. In a temperature range between 500 and 600 °C, the highest values for the $R_{TOC}$ were achieved. This can be explained due to an insufficient energy supply for oxidation at low temperatures, while at very high temperatures condensation reactions may have been favored [33]. Regarding the feed flow rate, the observed curvature in the plot indicates a decreasing trend in the $R_{TOC}$ at extremely high or low flow rates. The range of the feed flow rates that produced the most favorable results was between 10 and 18 mL/min, representing intermediate feed flow rates within the limits studied. Figure 2b shows the effects of the temperature and $H_2O_2$ concentration (related to the oxidation coefficient) in the $R_{TOC}$. The highest values for the $R_{TOC}$ were achieved when the oxidation coefficient was close to 2 and the temperature was around 600 °C. In the absence of $H_2O_2$, the effect of the temperature was not very pronounced on the removal of the TOC. Figure 2c shows the influence of the feed flow rate and $H_2O_2$ concentration on the $R_{TOC}$. The oxidation coefficient was the main factor influencing the $R_{TOC}$, with a higher reduction in the TOC being observed when the coefficient was greater than 1. The impact of the feed flow rate on the degradation process was minor; nonetheless, an increase in the

$R_{TOC}$ was noticeable under conditions of low feed flow rates. Based on the experimental results for the $R_{TOC}$ (%), a mathematical second order regression model was constructed, and the equation coefficients were estimated. The results suggest that both linear and quadratic effects related to the temperature, feed flow rate, and concentration of $H_2O_2$ were meaningful in the oxidation of the organic matter. By contrast, a synergic effect between the variables was not observed, as can be seen in Table 3, where the bidirectional parameters showed a *p*-valor greater than 0.01.

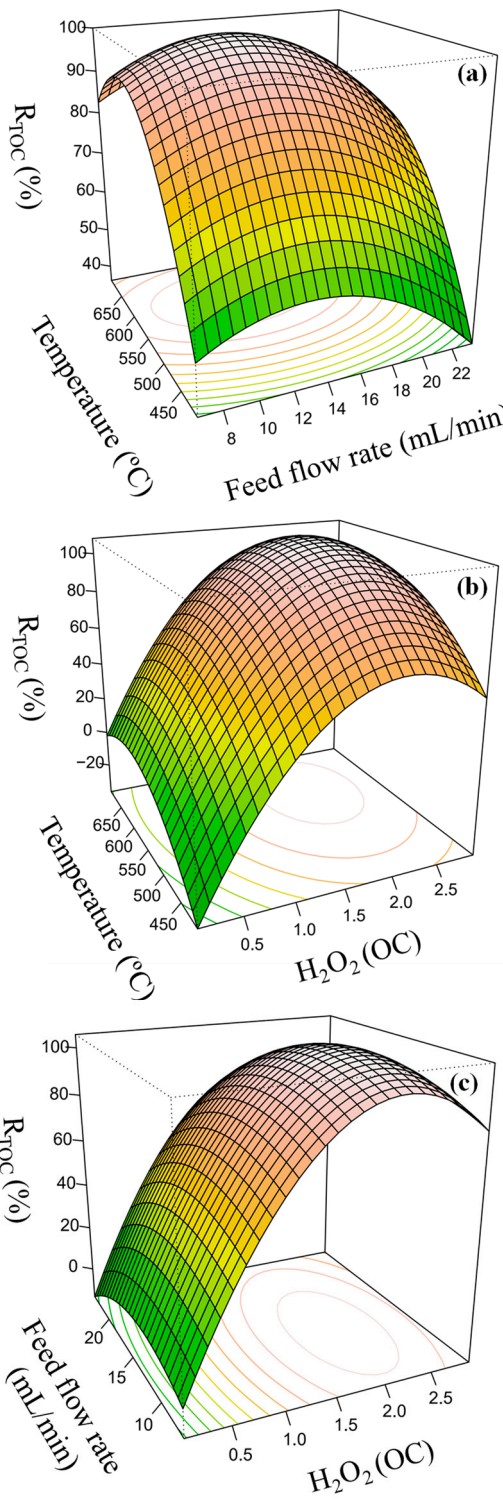

**Figure 2.** Results of the $R_{TOC}$ (%) as a function of (**a**) temperature vs. feed flow rate, (**b**) temperature vs. $H_2O_2$ concentration, and (**c**) feed flow rate vs. $H_2O_2$ concentration.

**Table 3.** Coefficient values of the regression model for the studied parameters.

| Response | Factor | Effect | Std. Error | t-Valor | Pr (> \|t\|) | [1] S. L. |
|---|---|---|---|---|---|---|
| [2]$R_{TOC}$ (%) | Mean | 96.264 | 3.283 | 29.321 | $1.9 \times 10^{-9}$ | *** |
| | T | 10.289 | 1.780 | 5.779 | $4.1 \times 10^{-4}$ | *** |
| | V | −3.529 | 1.780 | −1.982 | 0.082 | |
| | C | 24.619 | 1.780 | 13.828 | $7.2 \times 10^{-7}$ | *** |
| | T × V | −0.056 | 2.325 | −0.024 | 0.981 | |
| | T × C | −2.233 | 2.325 | −0.960 | 0.364 | |
| | V × C | −3.498 | 2.325 | −1.504 | 0.170 | |
| | $T^2$ | −8.577 | 1.851 | −4.631 | 0.001 | ** |
| | $V^2$ | −4.539 | 1.851 | −2.451 | 0.039 | * |
| | $C^2$ | −17.72 | 1.851 | −9.472 | $1.1 \times 10^{-5}$ | *** |

Note(s): [1] Significance level: 0 '***'; 0.001 '**'; 0.01 '*'; and 0.1 ' ' 1. [2] $R^2$: 0.9764; adjusted R: 0.9408.

The ANOVA table comprising the proposed model is reported in Table S3. The coefficient of determination for the equation under consideration was 0.8527, signifying that a substantial portion, specifically 85.27%, of the data variance can be accurately described by the mathematical model. It is noteworthy to highlight that these findings have attained statistical significance, as confirmed through a hypothesis test conducted at a confidence level of 90%, with a significance level established as $p \leq 0.1$. Moreover, it is worth noting that the residual analysis exhibited randomness and independence, with a mean value of zero and consistent variation.

After the evaluation of these parameters' effects, the reaction conditions were optimized. The result estimated by differential evolution showed that an $R_{TOC}$ of 99.8% could be achieved at a temperature of 600 °C, a feed flow rate of 13.3 mL/min, and an oxidation coefficient of 0.65. To validate the predicted result, an experimental test was performed at these optimized conditions. The $R_{TOC}$ achieved was 92.1%, indicating an error rate lower than 8%. This outcome further strengthens the validation of the mathematical model. Furthermore, the optimized conditions are in accordance with the response surface plots, where it was shown that the $R_{TOC}$ is favored by temperatures, feed flow rates, and oxidation coefficients that are in the intermediate conditions of the evaluated ranges.

### 3.2. Results of Chemical Characterization

3.2.1. Liquid Phase

Samples of the simulated wastewater containing antihypertensive and cardiovascular medications were characterized both before and after the supercritical water treatment. The results are shown in Table 4. The characterization results were used to evaluate the quality of the wastewater process treatment using supercritical water oxidation. In this regard, the physical–chemical parameters were compared to the concentration limits required by environmental regulatory agencies.

**Table 4.** Physicochemical characterization of simulated wastewater samples.

| Parameters | Sample | | | Environmental Regulations [3] | | | |
|---|---|---|---|---|---|---|---|
| | Untreated | Treated [1] | Uncertainty [2] | EEA [4] | USEPA [5] | CONAMA [6] | CODEGO [7] |
| pH | 5.64 | 6.87 | 0.01 | - | 6–9 | 5–9 | 6–9 |
| TOC | 332 ± 7 | 25 ± 6 | - | ≥95% | - | - | - |
| BOD | 422.0 | 13.0 | 0.145 | 300 | 53.0 | >60% | 500.0 |
| COD | 1044.30 | 40.20 | 0.06 | - | - | - | 1000.0 |
| Nitrate | 1.80 | 1.80 | 0.03 | - | - | - | - |
| Nitrite | <0.010 | 0.250 | 0.004 | - | - | - | - |
| Al | 0.2052 | 0.0349 | 0.0023 | | - | - | - |
| Ca | 2.174 | 0.003 | 0.003 | | - | - | - |
| Cu | 0.0089 | 0.0077 | 0.0007 | 1.0 | 1.0 | 0.5 | 0.24 |

**Table 4.** *Cont.*

| Parameters | Sample | | | Environmental Regulations [3] | | | |
| --- | --- | --- | --- | --- | --- | --- | --- |
| | Untreated | Treated [1] | Uncertainty [2] | EEA [4] | USEPA [5] | CONAMA [6] | CODEGO [7] |
| K | 5.303 | 5.044 | 0.004 | - | - | - | - |
| Mg | 1.665 | 0.049 | 0.001 | - | - | - | - |
| Mo | < | 0.012 | 0.003 | - | - | - | - |
| Na | 4.526 | 4.724 | 0.005 | - | - | - | - |
| Ni | 0.008 | 0.004 | 0.004 | 2.0 | - | 1.0 | 1.45 |
| P | 19.268 | < | 0.003 | - | - | - | - |
| S | 16.42870 | 1.27374 | 0.00021 | - | - | - | - |
| Se | 0.06190 | 0.06701 | 0.00006 | 0.3 | 0.3 | - | - |
| V | < | 0.03910 | 0.00003 | - | - | - | 0.06 |
| Zn | 0.096 | < | 0.006 | 5.0 | 5.0 | 2.0 | 0.42 |

Note(s): [1] Treatment conditions: temperature, feed flow rate, and $H_2O_2$ concentration were 600 °C, 13.3 mL/min, and 0.65, respectively. [2] Uncertainty = Expanded uncertainty (U) based on the combined standard uncertainty (confidence level of 95%; k = 2). [3] Regulated limit values are expressed in mg/L. [4] European Environment Agency (EEA) [34]. [5] United States Environmental Protection Agency (U.S. EPA) [35]. [6] Conselho Nacional do Meio Ambiente (CONAMA) [36]. [7] Companhia de Desenvolvimento Econômico de Goiás (CODEGO) [37].

The treatment process reduced the concentration of TOC from 332 to 25 mg/L, corresponding to an $R_{TOC}$ of 92.1%. Moreover, significant reductions of 97% and 96% were observed in the BOD and COD values, respectively. These results, obtained under optimized conditions, demonstrate the effectiveness of the treatment process in degrading organic matter. These findings were attained within a reaction time of approximately 2 min, underscoring the efficacy of the treatment process.

The treatment process had no impact on the nitrate concentration of both treated and untreated samples. On the other hand, the concentration of nitrite had a substantial increase of 25 times. This increase in nitrite levels may be attributed to the ionic dissociation of salts generated under supercritical conditions [38]. However, it is important to note that the relevant regulatory standards consulted do not establish specific limit values for the concentration of nitrite. The concentration of metals was also analyzed. The results showed that the concentrations decreased substantially, except for molybdenum, sodium, selenium, and vanadium, whose concentrations increased. This behavior can be attributed to the corrosion of the reactor wall alloy. Corrosion processes, which occur during the SCW oxidation treatment process, typically lead to an elevation in metal concentrations within the liquid phase [39]. In general, the physicochemical parameters of the treated samples met the limits required by the main water and effluent regulatory and control agencies.

The degradation of antihypertensive and cardiovascular medications contained in the aqueous solution was tracked by high-resolution mass spectrometry of both treated and untreated samples. In order to determine the molecular formulae of the compounds that were present in the aqueous solution, the presence of some atoms in the following range was taken into account: C (6–28), 13C (0–1), H (8–50), O (1–15), N (0–6), 37Cl (0–1), Cl (0–1), S (0–2), and Na (0–1). The results are shown in Figure 3:

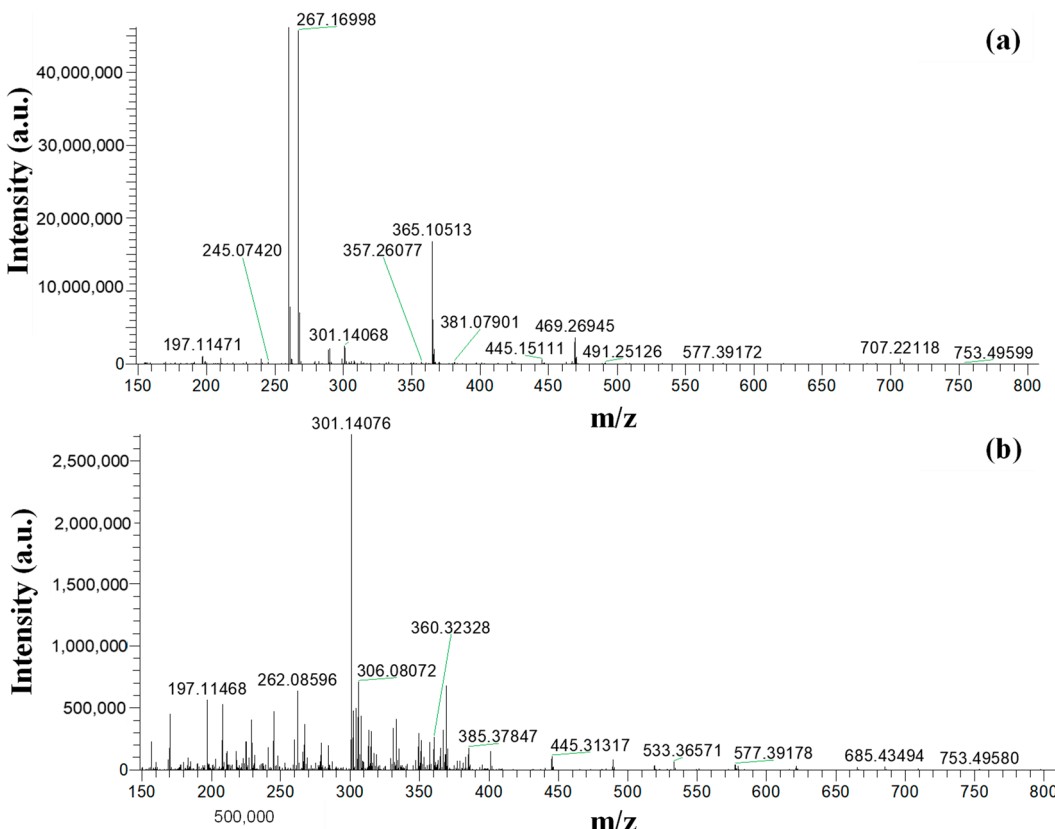

**Figure 3.** Mass spectra of (**a**) untreated and (**b**) treated wastewater solutions using the SCW process.

The treatment proved to be able to break multiple molecules, as evidenced by the presence of several peaks in a range of $m/z$ from 150 to 400 (Figure 3b). Although several signals were detected, they did not have their molecular formula attributed, due to the complexity of the samples (including the presence of pharmaceutical excipients).

By way of comparison with other technologies of treatment, Golovko et al. (2014) studied the removal and seasonal variability of analgesics/anti-inflammatory and anti-hypertensive/cardiovascular pharmaceuticals using UV filters in a wastewater treatment plant in the Czech Republic. The seasonal removal efficiency of 16 pharmaceuticals and personal care products was monitored, and, in most cases, the elimination of the substances was incomplete. Overall removal rates varied strongly from 38% to 100% [40]. In a comprehensive review reported by Ruziwa et al. (2023) of the photocatalytic degradation of pharmaceuticals in wastewater utilizing nano-enabled photocatalysts, the results indicated that the degradation rates achieved by several photocatalysts in pharmaceutical degradation varied from 70.5% to 99%. This range of degradation rates, when assessed in terms of the $R_{TOC}$, aligns closely with the findings in this study [41].

3.2.2. Gaseous Phase

The compounds generated in the gaseous phase at optimized treatment conditions were evaluated by gaseous chromatography. The obtained results can be seen in Figure 4. The products identified were hydrogen, carbon dioxide, carbon monoxide, and methane. The major component was carbon dioxide (97.9%) in a total gas flow rate of 26.3 mL/min. Hydrogen, methane, and carbon monoxide corresponded to 1.3%, 0.3%, and 0.5%, respectively.

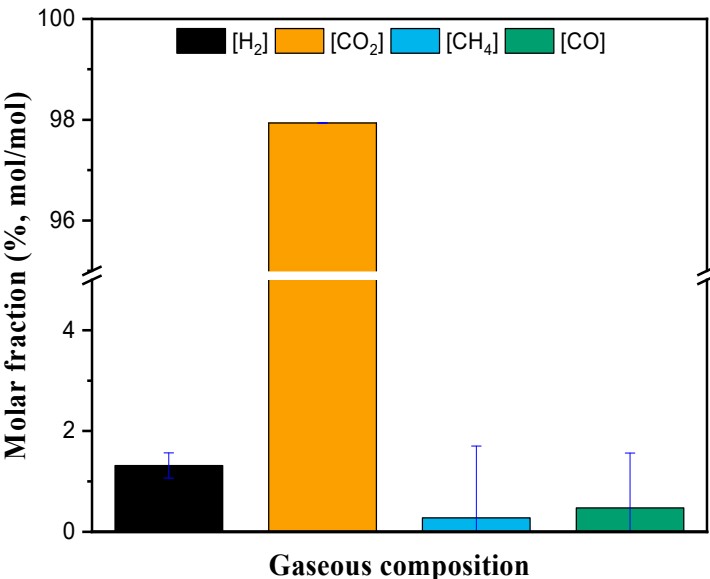

**Figure 4.** Gaseous composition observed during the SCW treatment of simulated wastewater samples.

The hypothesis that explains this performance, which supports the production of $CO_2$, suggests that the higher concentration of highly oxygenated molecules in the feed solution, along with the presence of $H_2O_2$, contributed to the increase in the production of $CO_2$. Moreover, C2-type gases were not found. This can be explained by the extremely oxidative aqueous medium at supercritical conditions, such as the presence of HO• radicals, which lead to the decomposition of organic compounds into lower molecular weight molecules (Equations (8)–(10)) and then the oxidization of these molecules into $CO_2$ and $H_2O$.

$$RH + HO\bullet \longrightarrow R\bullet + H_2O \tag{8}$$

$$R\bullet + O_2 \longrightarrow ROO\bullet \tag{9}$$

$$ROO\bullet + RH \longrightarrow ROOH + R\bullet \tag{10}$$

The application of the SCW process for the treatment of aqueous solutions contaminated with antihypertensive and cardiovascular molecules has shown efficient results of degradation through the SCW treatment. Regarding the gaseous products, the treatment process predominantly led to a complete degradation, with carbon dioxide being the primary gas formed. Since the main gaseous product is not sustainable, it is recommended to implement a method for capturing $CO_2$ in conjunction with the treatment process. One example is carbon fixation through the utilization of anaerobic ammonium oxidation (anammox) microorganisms, as reported by Wang, Han, and Zhang (2019) [41].

### 3.3. Results of Thermodynamic Equilibrium

The simulation of the thermodynamic equilibrium was used to assess the behavior of the reaction process. The main products predicted by the thermodynamic model were gaseous compounds, such as $CO_2$, $CH_4$, $H_2$, and $N_2$. Figure 5 presents the Spearman correlation matrix between the variables verified throughout the SCW treatment process. As an initial observation, it was noted that hydrogen formation has a negative correlation with the increasing ratio of pharmaceutical residues to water in the process feed. This result is justified by the fact that reducing the amount of water in the process feed hinders the water displacement reactions, thereby forming smaller quantities of hydrogen. Hydrogen formation also exhibits a negative correlation with the pressure, indicating that pressure increments can discourage hydrogen formation. This result is expected according to Le Chatelier's principle. For the same reason, the pressure shows a positive correlation with methane formation [30].

| | Temperature (°C) | Pressure (bar) | Phamaceutical Waste/$H_2O_2$ | Phamaceutical Waste/$H_2O$ | Hydrogen (% wt) | Methane (% wt) | Carbon Dioxide (% wt) | Nitrogen (% wt) | | |
|---|---|---|---|---|---|---|---|---|---|---|
| **Temperature (°C)** | 1.00 | 0.00 | −0.26 | −0.26 | 0.92 | −0.78 | 0.52 | −0.72 | | 1.00 |
| **Pressure (bar)** | 0.00 | 1.00 | 0.00 | 0.00 | −0.21 | 0.14 | −0.04 | 0.09 | | 0.71 |
| **Phamaceutical Waste/$H_2O_2$** | −0.26 | 0.00 | 1.00 | 1.00 | −0.37 | 0.62 | −0.23 | 0.44 | | 0.43 |
| **Phamaceutical Waste/$H_2O$** | −0.26 | 0.00 | 1.00 | 1.00 | −0.38 | 0.61 | −0.21 | 0.42 | | 0.14 |
| **Hydrogen (% wt)** | 0.92 | −0.21 | −0.37 | −0.38 | 1.00 | −0.89 | 0.60 | −0.82 | | −0.14 |
| **Methane (% wt)** | −0.78 | 0.14 | 0.62 | 0.61 | −0.89 | 1.00 | −0.80 | 0.95 | | −0.43 |
| **Carbon Dioxide (% wt)** | 0.52 | −0.04 | −0.23 | −0.21 | 0.60 | −0.80 | 1.00 | −0.94 | | −0.71 |
| **Nitrogen (% wt)** | −0.72 | 0.09 | 0.44 | 0.42 | −0.82 | 0.95 | −0.94 | 1.00 | | −1.00 |

**Figure 5.** Spearman correlations for the SCW treatment of pharmaceutical wastewater.

In addition to the variables already examined, the temperature showed a strong positive correlation with hydrogen formation, and this is related to the fact that the hydrogen formation reactions involved in the SCW treatment are predominantly endothermic. Therefore, temperature increments favor hydrogen formation and hinder the formation of components such as methane, carbon monoxide, and nitrogen for the same reason. Similar results to these presented for the behavior of hydrogen formation were observed by Mitoura et al. (2022) when investigating reactions in a supercritical state [30].

Following the behavior of the percentage of hydrogen formed throughout the process, the mass percentage of carbon dioxide shows a positive correlation only with the temperature. Thus, conditions that favor hydrogen formation also favor carbon dioxide formation. The excess carbon dioxide formation behavior was predicted by Kenneth and Savage (2005) when they studied the oxidation of methylamine in supercritical water [42].

In summary, temperature increments are expected to favor the formation of hydrogen and carbon dioxide while minimizing the formation of the other components ($N_2$ and $CH_4$). Increases in pharmaceutical waste residues should maximize the formation of carbon monoxide, nitrogen, and methane, while minimizing the formation of hydrogen and carbon dioxide. The formation profile of these compounds, as a function of the temperature and system pressure, is shown in Figure 6. To perform this analysis, the molar ratios of pharmaceutical waste/ $H_2O$ and the $H_2O_2$ concentration were kept constant at 12.5 and 10.0 wt%, respectively.

When analyzing the data presented in Figure 6c, it becomes evident that, within the specified pressure and temperature range relevant to this reaction, the predominant gaseous product generated was carbon dioxide ($CO_2$). This outcome is in accordance with the high molar ratio of carbon to hydrogen (C/H) in the feed stream, which was approximately 0.78. This finding aligns with expectations, especially in contrast to lignocellulosic materials, which have been previously reported as proficient hydrogen producers in SCW treatments, displaying C/H molar ratios of the order of 0.5, as indicated by Ren and colleagues (2022) [42].

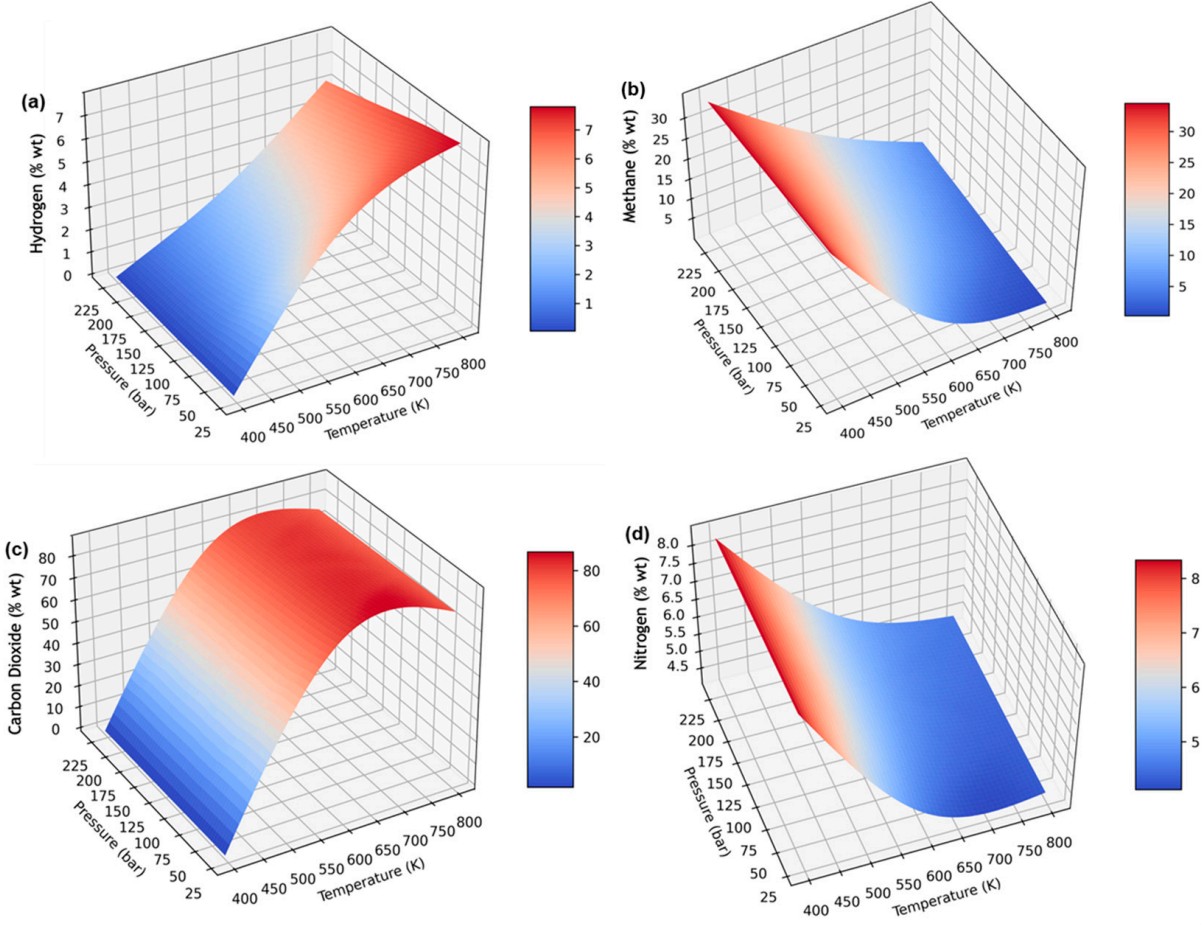

**Figure 6.** Pressure and temperature effects on the formation of the main reaction products during SCW treatment of the pharmaceutical waste system: (**a**) hydrogen; (**b**) methane; (**c**) carbon dioxide; and (**d**) nitrogen.

The high C/H molar ratio is also responsible for high $CH_4$ production, especially in low temperature conditions. These results can be visualized by observing Figure 6b. The highest $H_2$ productions of around 6% (considering the total mass of the system on a dry basis) were observed in the high temperature region, as depicted in Figure 6a. In general, the pressure had a low influence on the formation of gaseous compounds in this system, within the tested range, a behavior that was observed for all components. The main nitrogen degradation product that emerged from the simulations was nitrogen gas ($N_2$), as shown in Figure 6d. On a dry basis, the observed quantities were approximately 8.0% and were consistently maintained across the entire range of the pressure and temperature conditions employed in the simulation. In general, the formation of $HNO_3$, $HNO_2$, $NO_2$, $NO_3$, and $NH_3$ was not observed. Only small amounts of $NH_3$ (around $10^{-5}$ mols) were formed at low temperature conditions.

As the pressure entailed a secondary effect when compared with others, a more detailed study of the reaction behavior was performed, considering the molar feed ratio and the temperature as analysis variables. The results of this analysis are presented in Figure 7 in the form of contour lines. For this analysis, the pressure and the $H_2O_2$ concentration were kept constant at 230 bar and 10 wt%, respectively.

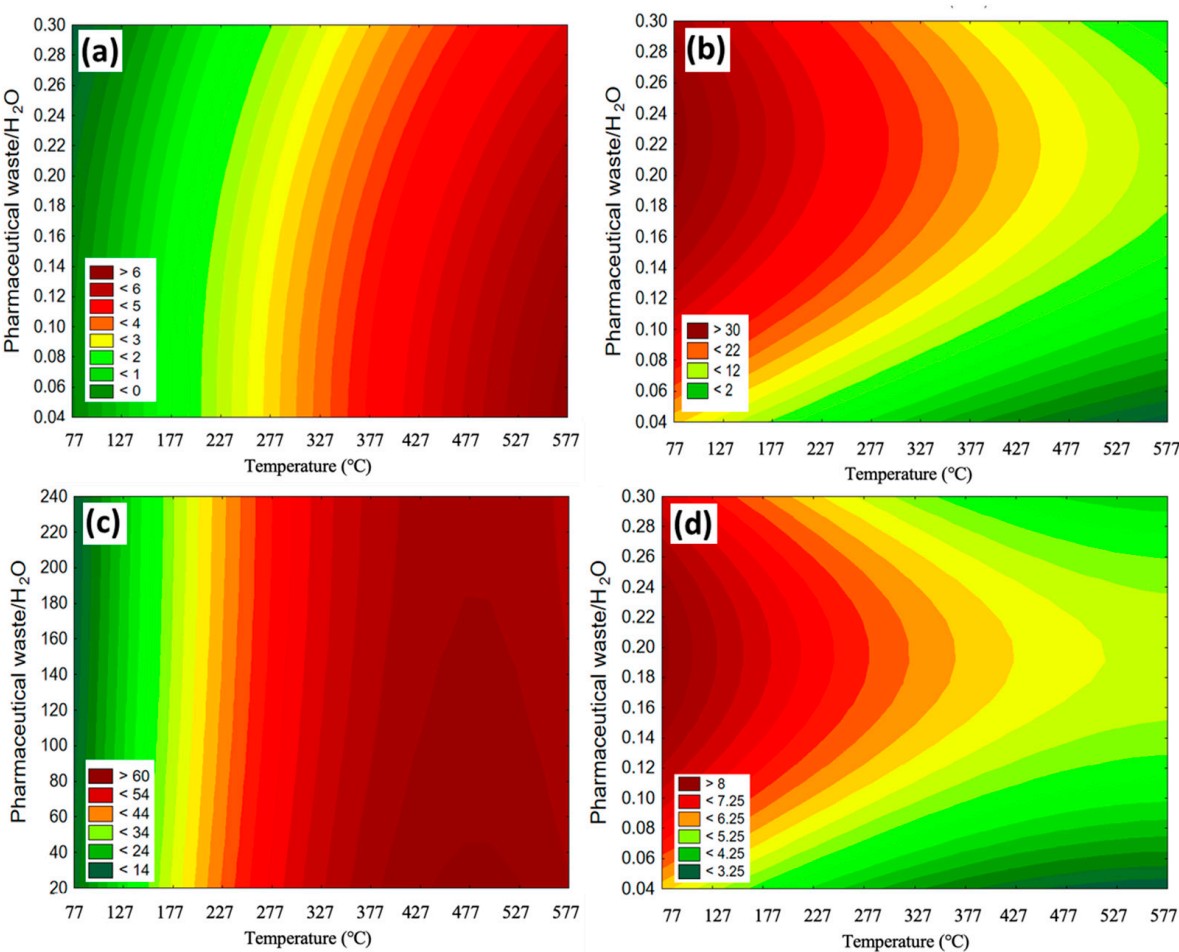

**Figure 7.** Combined effects of the pharmaceutical wastewater/$H_2O$ ratio and temperature on the formation of the main products: (**a**) hydrogen; (**b**) methane; (**c**) carbon dioxide; and (**d**) nitrogen.

The results showed that $CO_2$ remained as the main gaseous product formed within the entire range of temperature and the feed solution composition of pharmaceuticals studied in this work, around 60% of the gaseous product on a dry basis is composed of $CO_2$ (see Figure 7c). Hydrogen continued to be produced in small quantities (less than 7% for the entire range studied), as shown in Figure 7a. Another important effect observed was that the increase in the molar ratio of pharmaceutical residues in the feed solution was responsible for reducing the molar proportion of hydrogen in the gaseous product formed. This result was anticipated based on the molecular analysis of the treated effluent, which revealed the high carbon–hydrogen (C/H) molar ratio of the material. Moreover, the main nitrogen-derived product formed in the system was $N_2$, with proportions of around 8.0% of this compound in the dry product stream being observed at low temperatures (Figure 7d). The presence of a substantial proportion of methane ($CH_4$) in the gaseous products was noted, particularly in scenarios characterized by lower temperatures within the feed stream. This phenomenon was also reported in prior studies, such as in the study of Barros et al. (2022) about the SCWG of black liquor, and in the study of Ren (2022) about the SCW partial oxidation of ethanol [31,42].

## 4. Conclusions

The SCW processes applied to the treatment of wastewater contaminated with antihypertensive and cardiovascular medications showed promising results for the degradation of these molecules, which are considered emergent pollutants. Under optimal conditions of temperature, feed flow rate, and $H_2O_2$—600 °C, 13.3 mL/min, and 0.65,

respectively—the treatment achieved a TOC reduction of 92.1%. The organic matter was almost completely converted in the gaseous phase, where the main compounds determined were $CO_2$ (~98%) and $H_2$ (~1.3%). In the liquid phase, the findings indicated that when subjected to the SCW treatment, only atenolol, propranolol, and trimetazidine among the examined drugs—namely atenolol, captopril, propranolol hydrochloride, diosmin, hesperidin, losartan potassium, hydrochlorothiazide, and trimetazidine—did not undergo full degradation. Thus, further investigation to enhance the effectiveness of the treatment process is necessary, possibly through increasing the reaction time and/or reactor volume, as well as adding oxidant agents. Moreover, the incorporation of complementary treatment methods, such as adsorption, membrane filtration, and others, could be assessed. Despite this, the treated solution met with some of the main regulations of wastewater disposal, highlighting the benefits of the treatment process employed. Furthermore, the meticulous thermodynamic simulations provided a better understanding of the process behavior and contributed to a deeper knowledge of the process dynamics, particularly through the alignment of theoretical and experimental results. This synergy has informed discussions about enhancing the comprehension and development of the application of SCW processes in wastewater treatment. The treatment process proved to be fast and effective for the degradation of molecules.

**Supplementary Materials:** The following supporting information can be downloaded at: https://www.mdpi.com/article/10.3390/w16010125/s1. Detailed description of the physical-chemical characterization [26]; Table S1: Considered compounds during simulations and their thermodynamic properties [32]; Table S2: Operational conditions used in the SCW thermodynamic analysis via Gibbs energy minimization [30]; Table S3: Results of ANOVA table.

**Author Contributions:** I.M.D.: methodology, formal analysis, data curation, and writing—original draft. L.C.M.: validation, investigation, formal analysis, and writing. G.B.M.D.S.: validation, investigation, formal analysis and review, and editing. J.M.A.-P.: supervision, review, and editing. L.C.-F.: supervision, project administration, review, and editing. C.G.A.: supervision, project administration, and funding acquisition. J.M.D.S.-J., A.C.D.D.F. and R.G.: thermodynamic analysis of process and funding acquisition—review and editing. All authors have read and agreed to the published version of the manuscript.

**Funding:** Conselho Nacional de Desenvolvimento Científico e Tecnológico—CNPq (grants #407158/2013-8, #407158/2013-8, and #407158/2013-8) and Coordenação de Aperfeiçoamento de Pessoal de Nível Superior—CAPES—Brasil—Finance Code 001.

**Data Availability Statement:** All data are in the manuscript and supplementary material.

**Acknowledgments:** The authors gratefully acknowledge the financial support from Conselho Nacional de Desenvolvimento Científico e Tecnológico—CNPq (grants 407158/2013-8, 431642/2016-8, and 405851/2022-7). This study was financed in part by the Coordenação de Aperfeiçoamento de Pessoal de Nível Superior, Brazil—CAPES (Finance Code 001).

**Conflicts of Interest:** The authors declare that they have no known competing financial interests or personal relationships that could have appeared to influence the work reported in this paper.

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
