# Peer review of "Treatment of Antihypertensive and Cardiovascular Drugs in Supercritical Water: An Experimental and Modeled Approach"

_water, doi:10.3390/w16010125_

Round 1

Reviewer 1 Report

Comments and Suggestions for Authors

Dear Authors,

I respect the comprehensive examination of both liquid and gaseous phase outputs, which showcases the effectiveness of the SCW oxidation treatment and guarantees adherence to regulatory rules, thereby enhancing the research's significance. After carefully reading your proposal, I see that some changes are needed to improve the consistency and clarity of the content.  You can find the modifications I suggest below.

Title:

The length of your title is excessive. It should be shortened to improve clarity and conciseness.

1.     Introduction:

The third could be the first paragraph because it mentions more general information on pharmaceutical molecules. The first and second paragraphs could be one paragraph explaining the focused drugs and their data concisely.

Could you please give more information about the study problem or gap that your method fills? It would be easier to understand what your study adds to the body of current literature if you provided more information on this.

In addition to explaining the study gap, could you also make the main point of your paper more apparent? Knowing the exact goals will help the reader understand the scope of your study better. Also, please explain why you selected this method over others. This would help people understand your choices and show that the method fits your study goals.

2.      Materials and Methods:

Line 115:  Could you add the information about H2O2 in a different paragraph? This might help make things more easily understandable and help you focus on this important part.

Line 120: You may write 332 mg/L instead of “332 mg of carbon per liter.”

Line 122: You might want to change the caption from "2.2. Plug flow reaction unit" to "Supercritical Water (SCW) Reactor."

In this part, it would be helpful to give more information about the reactor, like what kind it is and whether the flow is constant.

3.     Results and Discussion

Figure 1 uses the acronyms 'TC' and 'FC' without an explanation. Could you define these abbreviations?

In Figure 5:   Figures 5a) and c) appear to have an inverted temperature scale compared to b) and d). Please check and make any necessary corrections to ensure consistency throughout the figure.

General Comment

Its practicality, scalability costs, and limitations should be investigated to improve comprehension of the study's usefulness and real-world applicability. It would be beneficial to have a quick conversation about these topics, especially any implementation difficulties and financial concerns

Author Response

Reviewer #1:

General comment:

I respect the comprehensive examination of both liquid and gaseous phase outputs, which showcases the effectiveness of the SCW oxidation treatment and guarantees adherence to regulatory rules, thereby enhancing the research's significance. After carefully reading your proposal, I see that some changes are needed to improve the consistency and clarity of the content.  You can find the modifications I suggest below.

Answer: Thanks a lot for the positive assessment of our efforts. We value your feedback.

Specific comment:

  1. The length of your title is excessive. It should be shortened to improve clarity and conciseness.

Answer: Thanks for the suggestion. Title was rewrite as Treatment of antihypertensive and cardiovascular drugs in supercritical water: an experimental and modeled approach”.

  1. Introduction: The third could be the first paragraph because it mentions more general information on pharmaceutical molecules. The first and second paragraphs could be one paragraph explaining the focused drugs and their data concisely.

Answer: Sentences were ordered as suggested.

Pages 1 and 2 – Lines 42 – 72

“Several classes of pharmaceutical molecules have been determined in the environment. As a consequence, these substances are now classified as emerging contaminants, capable of persisting in various environmental matrices [1,2]. Consequently, wastewater treatments are essential to remove such potentially toxic compounds. However, traditional methods of treatments are not efficient since wastewater treatment plants were not initially designed to eliminate emerging pollutants [3]. Cardiovascular diseases and hypertension represent significant global health issues, making necessary the use of antihypertensive medications. These medications play a crucial role in reducing cardiovascular morbidity, and, consequently, decreasing mortality [4]. In contrast, the presence of such medications in water bodies has gain attention, due to their potential negative effects on aquatic organisms and the disruption of ecosystem equilibrium [5].

Antihypertensive and cardiovascular medication act on the primary issues of therapeutic interventions, which are stabilizing irregular cardiac rhythms and regulating blood pressure [6]. In 2004, hypertension was directly responsible for 12.8% of global deaths and it still is a major cardiovascular risk factor with an important impact on mortality [7,8]. Furthermore, according to the World Health Organization, approximately 1.5 billion people worldwide have hypertension, a number expected to rise, which has led to an ever-increasing in the prescription and consumption of antihypertensive drugs [9]. Given the chronic nature of these conditions, long-term medical treatment becomes a necessity. In the human body, most of these drugs undergo hepatic metabolism and renal excretion, resulting in the release of a considerable quantity of unchanged drugs or by-products into the environment [10]. The presence of antihypertensive drug molecules in municipal wastewater has been observed in several countries. Subedi and Kannan (2015) monitored two centralized wastewater treatment plants in the Albany area (New York, USA) and determined that the daily dose per thousand inhabitants for atenolol, propranolol, diltiazem, and verapamil were 316, 50.7, 45.8, and 30 mg, respectively. In the Tagus Estuary (Portugal), the occurrence of antihypertensive drugs including indapamide (1.09 – 4.67 ng/L), irbesartan (7.57 – 161.9 ng/L), losartan (1.52 – 64.7 ng/L), as well as β-blockers such as atenolol (0.49 – 0.49 ng/L), bisoprolol (0.02 – 4.66 ng/L), carvedilol (0.53 – 1.01 ng/L), and propranolol (0.02 – 1.89 ng/L) was also observed [12].”

  1. Could you please give more information about the study problem or gap that your method fills? It would be easier to understand what your study adds to the body of current literature if you provided more information on this.

Answer: A new sentence was added to address this point.

Pages 2 – Lines 73 – 84

“The effectiveness/efficiency of various treatment methods, such as adsorption, ion exchange, ozonation, membrane separation, electro-oxidative processes, and advanced oxidative processes, has been evaluated for the degradation or removal of contaminants. However, it is important to note that these methods, when employed individually, may not be sufficient to adequately degrade or remove these contaminants [5,13–15]. Considering the need to treat recalcitrant contaminants, it becomes necessary to enhance treatment processes to achieve effective results. In this regard, the application of the supercritical water (SCW) technology can be a promising alternative for treating wastewater containing cardiovascular and antihypertensive drug molecules. Unlike most of the conventional treatments, that are not enough efficient or may show limitations for large-scale applications, SCW treatment shows highly oxidative potential, which allows fast and efficient gasification of organic molecules.”

  1. In addition to explaining the study gap, could you also make the main point of your paper more apparent? Knowing the exact goals will help the reader understand the scope of your study better. Also, please explain why you selected this method over others. This would help people understand your choices and show that the method fits your study goals.

Answer: Thank you for the suggestion. The text was rewritten, clarifying gaps and objectives.

Pages 2 – Lines 94 – 109

“[…] Nonetheless, the use of SCW processes for the degradation of antihypertensive and cardiovascular molecules in water has not been investigated in depth as far as authors of this work are aware. Hence, certain aspects can be enhanced and thoroughly assessed, including drug degradation rates, degradability of total organic carbon and chemical oxygen demand, reaction time, and the characterization of the gas phase generated during the treatment process.

Herein, given these gaps, continuous flow supercritical water process was evaluated and applied to treat simulated wastewater containing a wide range of medications commonly used worldwide for the treatment of cardiovascular diseases and hypertension. The simultaneous degradation of eight target molecules was investigated, namely atenolol, captopril, propranolol hydrochloride, diosmin, hesperidin, losartan potassium, hydrochlorothiazide, and trimetazidine. Furthermore, the effect of treatment process parameters, such as temperature, H2O2 concentration, and feed flow rate was assessed, as well as the behavior of the treatment process elucidation through the utilization of a simulation model at the thermodynamic equilibrium of the system.”

  1. Materials and Methods: Line 115:  Could you add the information about H2O2 in a different paragraph? This might help make things more easily understandable and help you focus on this important part.

Answer: It was revised.

  1. Line 120: You may write 332 mg/L instead of “332 mg of carbon per liter.”

Answer: It was revised.

  1. Line 122: You might want to change the caption from "2.2. Plug flow reaction unit" to "Supercritical Water (SCW) Reactor." In this part, it would be helpful to give more information about the reactor, like what kind it is and whether the flow is constant.

Answer: Thanks. It was revised and a new sentence added.

Pages 3 – Lines 138 and 139

“The tubular reactor is made of Inconel (VRC 625) and the other components are made of stainless steel (SS 316L).”

  1. Results and Discussion: Figure 1 uses the acronyms 'TC' and 'FC' without an explanation. Could you define these abbreviations?

Answer: It was revised.

Pages 4 – Lines 133 and 134

“FC and TC corresponds to flow controller and temperature controller, respectively.”

  1. In Figure 5:   Figures 5a) and c) appear to have an inverted temperature scale compared to b) and d). Please check and make any necessary corrections to ensure consistency throughout the figure.

Answer: It was revised.

  1. Its practicality, scalability costs, and limitations should be investigated to improve comprehension of the study's usefulness and real-world applicability. It would be beneficial to have a quick conversation about these topics, especially any implementation difficulties and financial concerns.

Answer: To achieve successful scaling-up, further investigation and enhancements addressing challenges such as corrosion, salt precipitation, clogging, and high operational costs — commonly associated with supercritical water processes — are necessary. It is noteworthy, however, that several corporations, including General Atomic (U.S.), Foster Wheeler (U.S.), SRI International (U.S.), HydroProcessing Company (U.S.), Supercritical Fluids International (Ireland), Chematur Engineering AB (Sweden), and Hanwha (South Korea), have already commercialized supercritical water processes. This underscores the feasibility of large-scale operations based on supercritical water technology, as reported by Botelho and co-workers (2022). DOI: https://doi.org/10.1007/s11157-021-09601-0

Reviewer 2 Report

Comments and Suggestions for Authors

Dear Authors,

I am pleased to entertain a reconsideration of the article for publication upon addressing the following scientific concerns:

  1. The absence of adequate controls in the conducted research warrants attention.

  2. The effects of chemicals and temperature must be individually and collectively addressed, with separate consideration for ambient temperature and zero chemicals.

  3. The impracticality of implementing this technology in an industrial sewer treatment plant requires elaboration.

  4. A comparative analysis of your results against those obtained from alternative technologies is essential.

  5. The utilization of authentic sewer water in experiments is imperative, as it may reveal additional issues or contaminants in the final product.

  6. The experimental procedure's methodology lacks clarity and requires further elucidation.

Comments on the Quality of English Language

The English language is fine, minor editing is required. 

Author Response

Reviewer #2:

General comment:

I am pleased to entertain a reconsideration of the article for publication upon addressing the following scientific concerns:

Answer: Thank you for the consideration of our article. We carefully tried to address all points raised by the referee. However, the lack of details makes it difficult to provide an accurate response.

Specific comments:

  1. The absence of adequate controls in the conducted research warrants attention.

Answer: The suggestion lacks clarity, and unfortunately, we were unable to comprehend the intended meaning of the term "adequate control." Consequently, we have chosen to disregard this recommendation.

  1. The effects of chemicals and temperature must be individually and collectively addressed, with separate consideration for ambient temperature and zero chemicals.

Answer: The individual and collective effects of temperature and hydrogen peroxide addition were discussed in Page 9, Section 3.1, Figure 2 (b), lines 263 – 267, as follows:

“Figure 2b shows the effects of temperature and H2O2 concentration (related to the oxidation coefficient) in the RTOC. The highest values for RTOC were achieved when the oxidation coefficient was close to 2 and the temperature was around 600 °C. In the absence of H2O2, the effect of temperature was not very pronounced on the TOC removal.”

Additionally, the supercritical water treatment process was also carried out in the absence of an oxidizing agent (zero chemicals?), and the results as well as the discussion were provided in Page 9, Section 3.1, Figure 2 (c), lines 267 – 272, as follows.

“Figure 2c shows the influence of the feed flow rate and H2O2 concentration on the RTOC. The oxidation coefficient was the main factor influencing the RTOC, with higher TOC reduction being observed when the coefficient was greater than 1. The impact of feed flow rate on the degradation process was minor, nonetheless, an increase in the RTOC was noticeable under conditions of low feed flow rates.”

Finally, a “separate consideration for ambient temperature” would be impractical since the very process that we are proposing requires (demands!) water under conditions of temperature and pressure above its critical point.

  1. The impracticality of implementing this technology in an industrial sewer treatment plant requires elaboration.

Answer: The results showed that the supercritical water treatment process is effective for the degradation of both antihypertensive and cardiovascular drugs on a bench scale operation.

As mentioned previously, for a successful scaling-up, additional investigation and improvements regarding corrosion, salt precipitation, clogging issues and high operational costs, drawbacks that are often associated with supercritical water processes, are still necessary.

However, it is critical to point out that supercritical water processes have already been commercialized by several corporations like General Atomic (U.S.), Foster Wheeler (U.S.), SRI International (U.S.), HydroProcessing Company (U.S.), Supercritical Fluids International (Ireland), Chematur Engineering AB (Sweden) and Hanwha (South Korea), indicating the feasibility of large-scale operations based on supercritical water technology.

  1. A comparative analysis of your results against those obtained from alternative technologies is essential.

Answer: Thank you. We have incorporated additional comparative data from other studies to address the raised point. The improvements can be seen below:

Pages 12 – Lines 363 – 373

“[…]

By way of comparison with other technologies of treatment, Golovko and co-workers (2014) studied the removal and seasonal variability of analgesics/anti-inflammatory, anti-hypertensive/cardiovascular pharmaceuticals using UV filters in a wastewater treatment plant in Czech Republic. Seasonal removal efficiency of 16 pharmaceuticals and personal care products was monitored, and in most cases, elimination of the substances was incomplete, and overall removal rates varied strongly from 38 to 100%[40]. In a comprehensive review reported by Ruziwa and colleagues (2023) on the photocatalytic degradation of pharmaceuticals in wastewater utilizing nano-enabled photocatalysts, the results indicated that the degradation rates achieved by several photocatalysts in pharmaceutical degradation varied from 70.5% to 99%. This range of degradation rates, when assessed in terms of RTOC, aligns closely with the findings in this study [41].”

  1. The utilization of authentic sewer water in experiments is imperative, as it may reveal additional issues or contaminants in the final product.

Answer: We agree with the referee, but unfortunately, experimental tests with real samples containing antihypertensive and cardiovascular medications were not conducted. On the other hand, the treatment by supercritical water oxidation/gasification of authentic and simulated wastewater samples presents little or almost no distinction concerning optimal treatment conditions, which are preferably high temperatures and low feed flow rates (directly related to the reaction time). Previous studies reported by our group applying supercritical water treatment in the degradation of amoxicillin (https://doi.org/10.1016/j.watres.2023.119826), hormones (https://doi.org/10.1016/j.jece.2021.106095), and residues from biodiesel manufacture (https://doi.org/10.3390/w15234062) revealed that technology was effective in treating simulated and authentic samples of wastewater.

  1. The experimental procedure's methodology lacks clarity and requires further elucidation.

Answer: The experimental procedure’s methodology was structured into subsections. Details regarding the preparation of simulated wastewater used are described, and the apparatus has a schematic representation and component description. Furthermore, the parameters of all the reaction tests performed (18 tests) are shown in Table 1. Physical-chemical characterization carried out also were described and methods details can be consulted in Supplementary Material. Thermodynamics analysis procedures also were meticulously reported.

  1. The English language is fine, minor editing is required. 

Answer: Thank you. All manuscript was spell checked.

Round 2

Reviewer 2 Report

Comments and Suggestions for Authors

The authors have addressed the majority of the reviewer's comments.

Comments on the Quality of English Language

English language is OK.